# TOWARDS REPORTING BIAS IN VISUAL-LANGUAGE DATASETS: BIMODAL AUGMENTATION BY DECOUPLING OBJECT−ATTRIBUTE ASSOCIATION

## ABSTRACT

*Reporting bias* arises when people assume that some knowledge is universally understood and hence, do not necessitate explicit elaboration. In this paper, we focus on the wide existence of reporting bias in vision–language datasets, embodied as the object-attribute association, which can subsequentially degrade models trained on them. To mitigate this bias, we propose a bimodal augmentation (BiAug) approach through object–attribute decoupling to flexibly synthesize vision–language examples with a rich array of object–attribute pairing and construct cross-modal hard negatives. BiAug consists of three phases: (1) We employ large language models (LLMs) in conjunction with a object detector to detect and filter valid objects; (2) On the caption side, the LLM generates a detailed description for each object with each of the four preset attributes, and produces a corresponding hard negative counterpart; (3) On the image side, an inpainting model is used to modify the original object based on descriptions with different attributes. By doing so, the object-attribute association is decoupled. The synthesized examples explicitly complement omitted objects and attributes to learn, and the hard negative pairs steer the model to distinguish various attributes for an identical object. Our experiments demonstrated that BiAug excels not only in object-attribute understanding but also in improving the performance of zero-shot retrieval tasks on general benchmarks, such as MSCOCO and Flickr30K. BiAug refines the way of collecting text-image datasets. Mitigating the reporting bias helps models achieve a deeper understanding of vision–language phenomena, expanding beyond mere frequent patterns to encompass the richness and diversity of real-world scenarios. [1]

## 1 INTRODUCTION

*Reporting bias* denotes the inclination of individuals to under-report the information they have accessed (Gordon & Van Durme, 2013). This bias often arises when people assume that certain information, typically commonsense knowledge, is universally understood and, therefore, does not necessitate explicit elaboration, leading to the omission of some foundational details. Reporting bias rarely hinders human communication because individuals can infer the missing information from context and their own knowledge. However, it could be a crucial challenge in vision–language (VL) datasets because VL models do not inherently possess the ability to grasp commonsense knowledge, making them susceptible to misinterpretations when faced with reporting bias.

In standard VL datasets, images are accompanied by descriptive captions. Considering captions are typically collected by either automatic web crawling (Schuhmann et al., 2022), human annotating (Lin et al., 2014), or even generated by LLMs (Fan et al., 2023), the reporting bias issue would widely appear in existing large-scale VL datasets. For instance, Figure 1 (a) presents two examples highlighting reporting bias. The first example showcases two images both labeled with the caption *'A dog runs with a tennis ball in its mouth'*, omitting details such as the dog's color and whether the background is grass or snow. Similarly, the second example provides two images captioned as *'salmon dish on the table'*, but one displays sliced salmon and the other a whole fish. Despite each caption being accurate for its respective image, a VL model trained on such data may struggle to discern nuances like *black vs. brown dog*, *snow vs. grass*, or *sliced vs. salmon fish*. Hence, mitigating the reporting

---

[1]The codes will be publicly available after the paper is published.

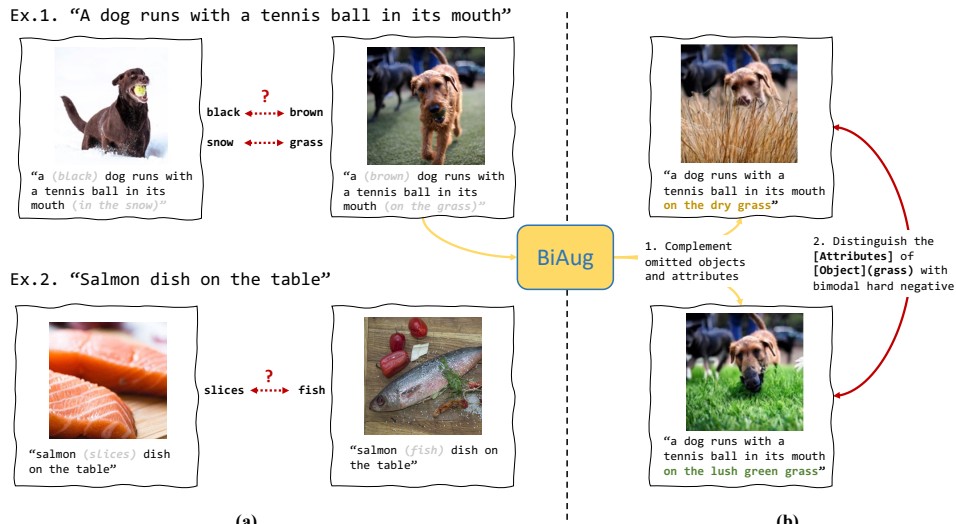

Figure 1: **(a)**: An illustration of reporting bias. *Gray texts* refer to the information that could be omitted. The given examples have identical captions, while the images have different objects (snow vs. grass), or the objects have different attributes (slices vs. fish). **(b)**: Bimodal augmentation (**BiAug**) not only complements omitted objects and attributes for both caption and images but, also constructs hard negative pairs steering the model to distinguish attributes.

bias in VL datasets is crucial for enhancing the performance of VL models trained on them. This need arises from several concerns: **(1):** Biased captions, which might be perceived as lacking objects or attributes, can be associated with multiple images that are dissimilar. Such imprecise pairings can compromise the training quality of VL models because they do not naturally have the capability to grasp commonsense knowledge to discern the difference. **(2):** Reporting bias skews the VL model towards frequently occurring patterns. For instance, with reporting bias, a search for *'a flag'* might predominantly yield images of a USA flag, ignoring the broader spectrum of flags. This bias hinders the model's efficacy in distinguishing nuanced object–attribute combinations.

We introduce a novel bimodal data augmentation framework, denoted as **BiAug**, that strategically disentangles object–attribute association for this problem. As demonstrated in Figure 1 (b), given a caption-image pairing, BiAug is designed to:

1. Synthesize **both new captions and corresponding images**. In this process, the caption's object receives additional descriptive detail, and the image's object undergoes a corresponding edit. Through the disentanglement of object–attribute association, BiAug crafts bimodal hard negative examples that emphasize a particular attribute.

2. Given that the object and attribute are decoupled, BiAug possesses the flexibility to produce samples with a **rich array of object–attribute pairings**. This feature helps diminish the over-representation of recurrent patterns, thus enabling the model to distinguish varying attributes for the same object.

We utilize BiAug to augment existing datasets and to evaluate BiAug by comparing models trained on the augmented dataset and the original source dataset, respectively. Our investigations span a variety of benchmarks. Primarily, VL models trained with BiAug consistently surpass baseline models on compositionality benchmarks. These benchmarks gauge a model's aptitude for grasping intricate commonsense knowledge. In addition, our trials on general text-image retrieval benchmarks also indicate that BiAug outperforms the baseline, which could be empirical evidence of mitigating the noise caused by reporting bias. BiAug refines the way of collecting text-image datasets. Mitigating the reporting bias ensures that models can achieve a deeper understanding of vision–language phenomena, expanding beyond mere frequent patterns to encompass the richness and diversity of real-world scenarios.

## 2    RELATED WORK

State-of-the-art vision–language representation learning methods (Radford et al., 2021; Jia et al., 2021; Yu et al., 2022; Li et al., 2023) are built upon contrastive language-image learning that pulls positive image-text pairs closer in a latent space while pushing negatives apart. The learning paradigm enables a broad understanding and generalization of various vision–language tasks, including zero-shot recognition, visual question answering (Li et al., 2022; 2023), etc. At the core of this success is the sheer scale of the web-scraped image-text pairs available for training, e.g., 400M pairs collected by CLIP (Radford et al., 2021) and 5B pairs collected by LAION (Schuhmann et al., 2022).

**Improving VL pertaining datasets.**    This, however, comes with several undesired properties underneath the data. Since the data is crawled from the Internet with minimal human effort, it is noise- and bias-prone because human tends to only report the contents of interest. As a result, a recent strand of research is devoted to cleaning and improving VL pretraining dataset quality. Radenovic et al. (2023) propose a series of methods of improving the LAION dataset, such that a ViT-Large CLIP model trained on 438M text-image pairs performs on par with a ViT-Huge CLIP model trained on 2B text-image pairs. Improving dataset quality is also critical for training diffusion models, e.g., LAION-Aesthetics is a subset of LAION-5B in which the text-image pairs are preferable to humans.

**Caption augmentation.**    A line of research has explored augmenting text-image datasets with synthetic captions. (Li et al., 2022) proposed a bootstrapping framework for both vision–language alignment and caption generation. (Santurkar et al., 2022) showed that training CLIP solely on synthetic captions generated by BLIP (Li et al., 2022) could even outperform a counterpart trained with web-crawled captions while a subsequent work by (Nguyen et al., 2023) further investigated the strategies to make the best use of both raw and synthetic captions. More relevant to this work, NegCLIP (Yuksekgonul et al., 2022) discussed the compositional performance of the vision–language model. However, they only considered augmenting captions while ignoring the images, making their method prominently different from our method.

**Image augmentation.**    Driven by the unprecedented success of text-to-image generation models (Rombach et al., 2022), several works (Sehwag et al., 2021; He et al., 2022; Azizi et al., 2023; Bansal & Grover, 2023) have demonstrated that synthetic data can boost the performance of image recognition. In the context of vision–language learning, StableRep (Tian et al., 2023) proposed using multiple images generated with the same caption as positive pairs for contrastive learning, showing promising results on image representation learning but not on text-image alignment. In contrast, BiAug augments the text-image datasets from *both caption and image perspectives*, and is capable of improving the vision–language alignment as well as eliminating the reporting bias.

To the best of our knowledge, BiAug is the first approach that considers the commonsense knowledge within VL datasets from the aspect of reporting bias. Utilizing advanced LLMs, which encompass extensive real-world knowledge (Petroni et al., 2019), BiAug highlights the commonsense knowledge of datasets through joint image and caption synthesis to mitigate the reporting bias problem.

## 3    BIAUG: BIMODAL AUGMENTATION FOR VL DATASETS

We discussed the difficulty for VL models to understand commonsense knowledge with existing text-image datasets in §1. To augment the source dataset with more explicit commonsense knowledge to learn using neural network models, we utilize an LLM coupled with visual tools to infuse commonsense knowledge in bimodal augmentation. In this section, we introduce the framework of BiAug, consisting of three phases: (I) LLM-aided object detection; (II) decoupling object–attribute association in captions; (III) editing images with produced commonsense knowledge and constructing hard negatives. We will introduce these three phases in detail in the following subsections. Please refer to §C for the specific instruction we use to prompt LLM.

### 3.1    LLM-AIDED OBJECT DETECTION

In §1, we highlighted the possibility of reporting bias leading to omitted objects. To mitigate this, we introduce an LLM-aided object detection technique to identify these overlooked objects. The process is elucidated through Steps 1 and 2 in Figure 2. **Step (1)**: given a caption from the source dataset, an LLM is employed to discern possible object candidates that could be visible in the described scene, as LLMs with commonsense reasoning ability are able to infer possible co-occurrent objects based

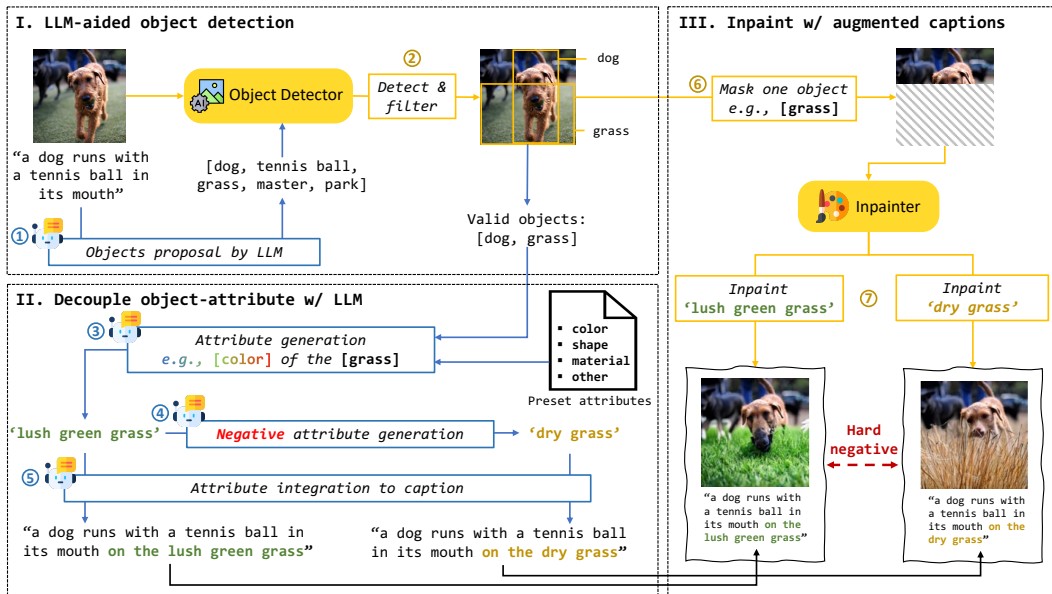

Figure 2: The illustration of BiAug with a detailed example, i.e., the "color" of "grass". The pipeline consists of three phases. The three phases can be split into seven steps, where steps highlighted in yellow are conducted with pre-trained visual foundation models and the ones in blue are conducted with LLM prompting. Details are introduced in §3.

solely on the context of the caption. The specific prompt utilized is provided in § C. **Step (2)**: both the possible object candidates and the image are processed using the GroundingDINO (Liu et al., 2023) to detect valid objects in the images. Object detector is utilized to verify the object proposals and retain the objects that are most likely present based on the visual cues. Any non-detected and low-quality objects are discarded in this step, details about the filtering strategies are introduced in §4.1. Notably, this LLM-aided method enables the extraction of objects *not explicitly referred to in the caption* through cross-modal validation.

## 3.2 DECOUPLE OBJECT−ATTRIBUTE ASSOCIATION

Upon identifying the visible objects within an image, we decouple the association between the identified objects and their attributes through Steps 3-5 in Figure 2. The prompt utilized during this phase is introduced in § C. **Step (3)**: the decouping is achieved by prompting the LLM to generate diverse attributive descriptions for a given object, focusing on four predefined commonsense attribute categories: *color*, *shape*, *material*, and *other*[2]. For instance, *'[color] of [grass]' → 'lush green grass'*. **Step (4)**: subsequently, the LLM is tasked with producing a counter-description for the identical object and attribute category, but a different attribute value, serving as a hard negative example to understand distinct attribute values. For instance, *'lush green grass' → 'dry grass'*. **Step (5)**: lastly, the LLM augments the initial caption with the generated attributive descriptions LLM separately integrates the attributes into the initial caption, forming a pair of hard-negative captions.

## 3.3 IMAGE SYNTHESIS AND HARD NEGATIVES CONSTRUCTION

To complement the information within augmented captions, we generate corresponding synthesized images, through steps 6 and 7 in Figure 2. **Step (6)**: first, we mask each detected object within the images according to the bounding box obtained in Step 2. **Step (7)**: using the stable diffusion inpainting model[3], we inpaint the masked area based on the generated descriptions in Steps 4 and 5 for the object. This approach allows us to produce images that match the augmented captions, offering variations in the attributive descriptions for the same object. Such diversity not only enhances the dataset but also addresses potential reporting bias.

---

[2]The 'other' category leverages the LLM's generalization capabilities to infer the most pertinent attributive description beyond the other specific three, based on the given caption.

[3]https://huggingface.co/stabilityai/stable-diffusion-2-inpainting

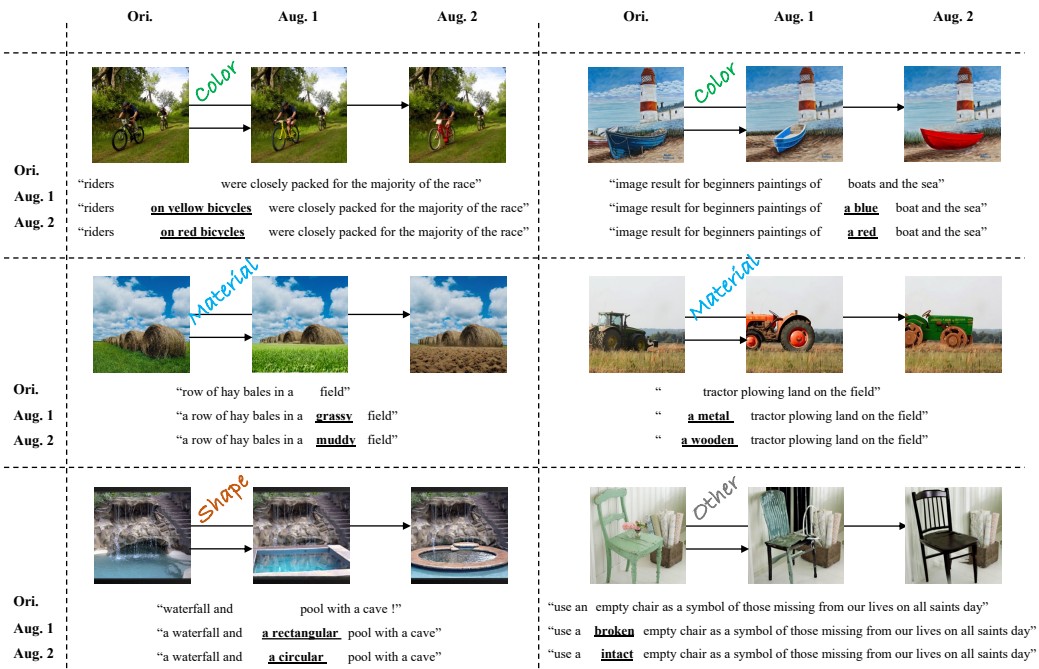

Figure 3: Sythesized examples by BiAug. As can be observed, BiAug successfully decouples the object–attribute pairs, e.g., "a blue boat" and "a red boat".

Furthermore, as shown in step 4 in Figure 2, BiAug provides hard negative descriptions for the same object within the identical attribute category. Consequently, these synthesized images function as mutual hard-negative images, and, combining with the hard-negative captions, form a hard-negative example of image-caption pairs. These bimodal hard negatives bolster the VL model's capacity to assimilate the provided commonsense knowledge more effectively. Finally, we combine the augmented and source datasets as the constructed dataset by BiAug.

## 4 SYNTHESIZED DATASET

### 4.1 IMPLEMENTATION DETAILS OF BIAUG

**Pretraiend LLMs and visual tools.** In §3, we introduce the use of LLMs and various visual tools for object extraction, object-attribute decoupling, and image generation. Specifically, we utilize gpt-3.5-turbo[4] as our LLM, GroundingDino[5] (Liu et al., 2023) as our object detector, and Stable-diffusion-inpainting[6] for image inpainting. It is important to note that our main contribution is the introduction of a unique bimodal augmentation framework that decouples object–attribute associations. While we have selected state-of-the-art public tools for our experiments in this paper, the proposed framework is expected to be increasingly effective, along with the fast development and evolution of these tools.

**Source datasets.** We extracted subsets of 40,000, 100,000, 200,000 and 300,000 examples from the Conceptual Caption 3M (CC3M)[7] dataset, labeled as 40K, 100K, 200K and 300K respectively. These subsets were subsequently augmented using BiAug. We choose to work on CC3M because CC3M captions contain hypernyms, to which the possible attributes are well defined.

**Filtering strategies.** BiAug generates new captions and images using established models and tools, which can sometimes introduce errors and noise. To ensure the quality of the augmented dataset, we employ various strategies to eliminate examples that are potentially corrupted: **(1)** To ensure the detected box is in a valid size for inpainting, we remove boxes that are smaller than 5% or larger than

---

[4]https://platform.openai.com/docs/guides/gpt/chat-completions-api
[5]https://github.com/IDEA-Research/GroundingDINO
[6]https://huggingface.co/stabilityai/stable-diffusion-2-inpainting
[7]https://ai.google.com/research/ConceptualCaptions/download

80% of the original image. **(2)** To maintain the integrity of synthesized images, objects that occupy over 70% of another object's area are removed. This minimizes disruptions during image generation; **(3)** To ensure that the extracted objects are distinctly visible in the images, a confidence threshold of greater than 0.9 is applied. It is noteworthy that strategy 1 and 2 is default features in the standard pipeline of BiAug, whereas strategy 3 corresponds to *filtering* as discussed later in this paper.

## 4.2 SYNTHESIZED EXAMPLES

Figure 3 shows synthesized samples by BiAug. BiAug extracts an object and decouples it with the associated attributes, and then flexibly generates images with diverse object–attribute pairing and hard negative counterparts, with the commonsense knowledge from LLMs.

## 4.3 STATISTICS OF DATASET

Table 1: Statistics of synthesized dataset. *: some of the examples in the source dataset are dropped due to issues such as overly long sequence.

| Source Dataset | 40K | 100K | 200K | 300K |
|---|---|---|---|---|
| # of source data * | 38,100 | 88,300 | 187,900 | 287,600 |
| # of extract objects | 39,640 | 91,472 | 194,571 | 297,567 |
| # of augmented examples | 122,026 | 280,764 | 599,860 | 921,874 |
| – after filtering | 77,700 | 178,746 | 381,275 | 586,278 |
| # of hard negative pairs | 61,013 | 140,376 | 299,910 | 460,908 |
| – after filtering | 30,325 | 69,748 | 148,690 | 228,605 |

Table 1 details the quantitative breakdown of datasets synthesized by BiAug. For every source dataset, we count the number of source examples, extracted and detected objects, augmented examples, and hard negatives. Through our augmentation pipeline, we extract and detect the same number of objects as the source examples. This process synthesizes approximately three times the number of original examples. When the confidence filtering strategy is applied, the number of synthesized examples drops to twice the original source examples. Overall, the ratio of synthesized examples to the original examples remains relatively consistent as the dataset size increases. Particularly, fewer hard negative pairs can be obtained when the filtering strategy is applied.

## 4.4 BREAKDOWN EVALUATION OF THE GENERATION QUALITY

To have an intuitive measurement for the performance of the language/visual foundation models we use, we conduct a human evaluation for every phase in BiAug. We randomly sampled 100 augmented examples containing the intermediate outputs of each step. Then we ask 3-5 annotators to answer five questions to the corresponding outputs of the foundation models. *(1) Correct object*: "In Steps 1 and 2, does the framework generate reasonable objects?" *(2) Implicit object*: "In Steps 1 and 2, are any of the valid objects not explicitly mentioned in the original caption?" *(3) Correct attribute*: "In Step 3, does the framework add a description for the correctly given attributes?" *(4) Valid negative*: "In Step 4, can the pair of attributive objects serve as hard negative example to each other?" *(5) Valid inpainting*: "In Step 7, does the model correctly inpatint the specified object in the masked area?" Table 2 shows the results for the above questions. It demonstrates that the foundation models can reach high correctness for object proposal and hard negative generation, which can meet two basic requirements to augment VL datasets. Wherein around 60% of examples are augmented by objects that are not explicitly mentioned in the original caption. However, the correctness of attributes and inpainting quality are relatively lower with a number around 60%. Although this could introduce noises in the augmented dataset, it can still provide knowledge to learn as we only make limited but specific modifications to the original dataset.

Table 2: Human evaluation of phases in BiAug. The annotator are required to answer whether the output of foudation models follow the instruction. The reported number is the average of all annotator.

| | Correct object | Implict object | Correct attribute | Valid negative | Valid inpainting |
|---|---|---|---|---|---|
| Yes | 93.75 | 60.75 | 59.66 | 96.00 | 63.00 |
| No | 4.50 | 36.50 | 22.33 | 1.00 | 21.33 |
| Unsure | 2.00 | 3.00 | 11.33 | 3.00 | 20.33 |

## 5 EXPERIMENTS

### 5.1 TRAINING DETAILS

**Source dataset, back-bone model and training details**    The baseline mode is labled as **CLIP-ft**, which is trained on the source datasets. The proposed method is labled as textbfBiAug, which train CLIP on the combination of BiAug dataset and the source dataset. We use ViT-B/16 as the back-bone VL model. We conduct a standard contrastive language-image training (Radford et al., 2021) to fine-tune the model. The bimodal hard negatives are added in the in-batch examples as negatives without particular handling. For both CLIP-ft and BiAug, we finetune the model starting from the OpenAI checkpoint[8]. The learning rate is set as a relatively small 1e-8, because the CLIP model is sensitive to fine-tuning. The batch size is 1024. We fine-tuned the model on augmented datasets for 5 epochs. Note that for the training of the source dataset, the amount of examples is less than that of our augmented dataset. For a fair comparison, we train the baseline model for more epochs to ensure it is trained with the same steps, e.g., if the size of BiAug dataset is 3 times that of the source dataset, we train the baseline model for $3 \times 5 = 15$ epochs.

### 5.2 EVALUATION ON OBJECT–ATTRIBUTE UNDERSTANDING DATASETS

**The choice of test datasets.**    According to our earlier discussion in §1, models trained on datasets characterized by reporting bias may exhibit a predilection for dominant object–attribute pairings, consequently introducing bias into subsequent tasks. In order to assess the capacity of the BiAug to discern objects and attributes independently instead of perceiving them as inseparable associations, we evaluate models on object–attribute comprehension. To this end, we have chosen to employ test datasets that are designed to probe into the challenging realm of object–attribute understanding because accurate attribution of different objects is a prerequisite for successful compositionality.

Taking an example from Yuksekgonul et al. (2022):

1. *the paved road and the white house.*

2. *the white road and the paved house.*

compositionality tasks require accurately attributing "white" to "a house" and "paved" to "a road", and distinguishing the sentences 1 and 2. This line of tasks matches well with the reporting bias problem in VL datasets: the association of object–attribute pairs. As a result, we evaluate the trained models on the ARO[9] benchmark containing four subsets of probing compositionality. We narrowed down our focus on subsets that are relevant to the object–attribute association problem based on the way of construction of these datasets.

**ARO.**    The Attribution, Relation, and Order (ARO; Yuksekgonul et al. (2022)) benchmark probes VL models's ability on compositionality. It consists of approximately 50k test examples derived from Visual Genome (VG), MSCOCO, and Flickr30K. This benchmark requires the model to choose the correct caption from a group of synthesized hard negative ones. Figure 4 compares BiAug, CLIP-ft, and the original CLIP without further fine-tuning. BiAug clearly outperforms the baselines on all subsets, and the improvements increase along with the availability of more augmented data.

**Evaluation on VG-Relation and VG-Attribute**    In addition to the overall advantage of BiAug, we also observe varying trends in dataset size on different subsets. VG-Relation and VG-Attribute construct hard negative testing captions by swapping the object phrase itself and attributive phrases of a pair of objects in the caption, respectively, which is more relevant to our evaluation of object–attribute association. Figure 4a and 4b show the results on VG-Relation and VG-Attribute. We can observe that BiAug gets improved with more training examples. However, CLIP-ft, the baseline, does not improve and even degrades with larger datasets. This indicates that BiAug provides more diverse examples that help the object–attribute understanding compared with the source dataset.

**Evaluation on Flicker30K-Order and COCO-Order**    On the other hand, the subsets of Flickr30K-Order and COCO-Order construct hard negative testing captions through shuffling the adjective/noun, unigrams, or trigrams in the captions, which is also a type of compositionality task, but less relevant to the object–attribute problem we focus on. The results are shown in Figure 4c and 4d. CLIP-ft fails on these two subsets with an obvious degradation with more training data, even worse than CLIP

---

[8]https://huggingface.co/openai/clip-vit-base-patch16
[9]https://github.com/mertyg/vision-language-models-are-bows

without fine-tuning. The possible reason for this phenomenon is that VL model could perform like bag-of-words and fail to identify different orders of words (Yuksekgonul et al., 2022). In contrast, BiAug performs more stably, indicating that the training data synthesized by BiAug describes the compositional information better than the original examples. Different from the performance on VG subsets, BiAug still cannot be further improved with the increasing size of datasets, which could be caused by the out-of-domain examples or the irrelevant problem definition.

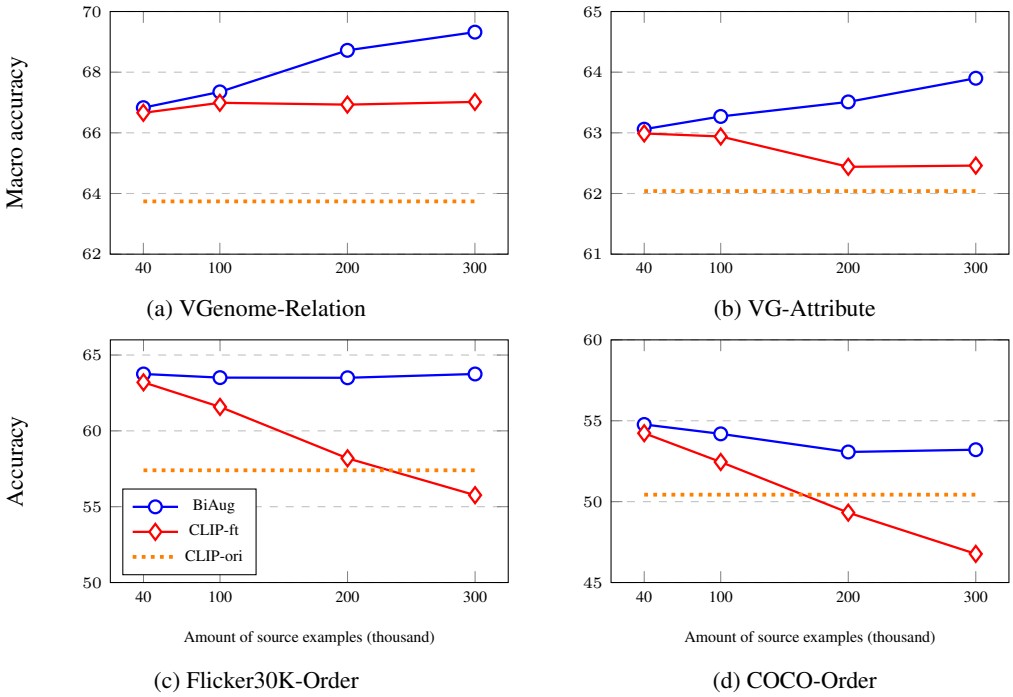

(a) VGenome-Relation

(b) VG-Attribute

(c) Flicker30K-Order

(d) COCO-Order

Figure 4: Comparison of BiAug, CLIP fine-tuning on the source dataset, and CLIP without fine-tuning on ARO benchmark. The x-asis refers to the size of source dataset (thousand). One each subset, the trend of performance on the dataset size is demonstrated. Particularly, VG-Relation and VG-Attribute construct the testing examples by swapping object and attributive words, while Flicker30K-Order and COCO-Order just shuffle the various words in the caption (Yuksekgonul et al., 2022).

## 5.3 EVALUATION ON GENERAL VISION–LANGUAGE RETRIEVAL DATASETS

§5.2 demonstrates the advantage of BiAug of improving CLIP's performance on object–attribute understanding. In this section, we also evaluate BiAug on two common benchmarks of retrieval, MSCOCO (Chen et al., 2015) and Flickr30K (Plummer et al., 2015). MSCOCO and Flickr30K are not designed to test the object–attribute understanding, but this evaluation can also verify if BiAug synthesizes better training examples by complementing the omitted objects and attributes from both visual and language modalities.

Table 3 compares model performances on the Karpathy test split (Karpathy & Fei-Fei, 2015) of MSCOCO[10] and Flickr30K[11]. In image retrieval, e.g., ImageAt1, the model input is a caption while the expected output is the corresponding image. In text retrieval, e.g., TextAt1, the model input is an image, while the expected output is the corresponding caption. In general, introducing BiAug leads to positive impacts on the retrieval results. This confirms that the applicability of BiAug is not limited to object–attribute understanding but could also be beneficial to general retrieval tasks.

## 5.4 SETTING DECISION

This section checks the impact of various settings within our approach. We present the experimental results for multiple variants of BiAug in Figure 5. To ensure fair comparisons, adjustments in

---

[10]https://paperswithcode.com/sota/cross-modal-retrieval-on-coco-2014

[11]https://paperswithcode.com/sota/cross-modal-retrieval-on-flickr30k

Table 3: Retrieval results on MSCOCO and Flickr30K. Image @K denotes the image retrieval with recall@K. Text @K denotes the text retrieval recall@K.

| Method | Image @1 | Image @5 | Image @10 | Text @1 | Text @5 | Text @10 |
|--------|----------|----------|-----------|---------|---------|----------|
| MSCOCO | | | | | | |
| CLIP | 33.07 | 58.41 | 68.98 | 52.38 | 76.72 | 84.60 |
| CLIP-ft | 34.59 | 59.84 | 70.23 | 54.78 | 78.08 | 85.34 |
| BiAug | **35.60** | **60.57** | **70.68** | **55.46** | **78.78** | **86.20** |
| Flickr30K | | | | | | |
| CLIP | 62.08 | 85.58 | 91.78 | 81.90 | 96.20 | **98.80** |
| CLIP-ft | 64.72 | 86.94 | 92.04 | 82.00 | 97.00 | 98.70 |
| BiAug | **65.36** | **87.26** | **92.76** | **82.20** | **97.10** | **98.80** |

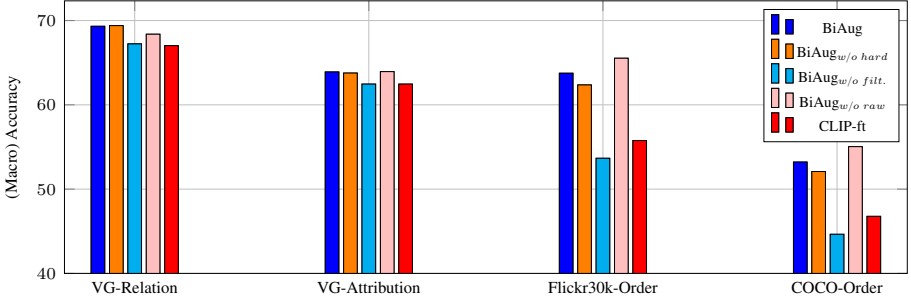

Figure 5: Ablation study of BiAug. $BiAug_{w/o\ hard}$ denotes training with augmented datasets while hard negative examples are not applied; $BiAug_{w/o\ filt.}$ denotes training on augmented datasets that are not filtered by strategies introduced in §4.1. $BiAug_{w/o\ raw}$ denotes training on augmented datasets where the source datasets are not included. The source dataset size is 300K.

training epochs are made to keep training steps comparable across these different variants, as they include different amounts of examples. Our results reveal that BiAug consistently outperforms the baseline across most scenarios. Removing hard negatives during training is associated with a decline in performance metrics for Flickr30K-Order and COCO-Order datasets, underscoring the valuable contribution of bimodal hard negatives in enhancing our understanding of compositionality. Notably, the adoption of filtering strategies emerges as a critical element, as their absence results in a notable degradation in performance across all subsets, even surpassing the baseline. This decline can be attributed to the utilization of existing tools within our approach, which may introduce noisy data and consequent error accumulation. Furthermore, the removal of source examples from augmented datasets yields an improvement in performance for Flickr30K-Order and COCO-Order datasets, underscoring the potential of our augmented datasets in enhancing VL models' grasp of object–attribute relationships and broader compositionality principles. It is important to note that, in the standard setting, we retain the raw examples to ensure greater diversity and maintain consistent performance across a range of potential downstream tasks.

## 6   CONCLUSION

This paper has extensively studied the problem of reporting bias, a crucial issue in large-scale text-image datasets. Our work sheds light on the challenges posed by reporting bias for vision–language models, emphasizing the deleterious effects of this bias on the VL model's ability to capture common-sense knowledge and the dominance of frequent patterns. As a solution, we introduced the bimodal data augmentation (BiAug) framework. BiAug allows for the synthesis of both new images and captions with enhanced object–attribute descriptions, by decoupling object–attribute associations to mitigate the limitations of reporting bias. Our experimental evaluations on various benchmarks showcase the significant advantages of BiAug. The framework not only strengthens the model's performance on compositionality tasks but also on standard text-image retrieval benchmarks. We believe that our work serves as a stepping stone towards refining the way of collecting text-image datasets. Mitigating the reporting bias ensures that models can achieve a deeper understanding of vision–language phenomena, expanding beyond mere frequent patterns to encompass the richness and diversity of real-world scenarios.

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

APPENDIX

## A  LIMITATION

The cost of data synthesis is expensive as we use large models like LLMs and stable diffusion. In this paper we focus on verifying the effectiveness of BiAug. The cost problem can be mitigated in the future work through several strategies like local deployment and parallel processing. Although we use state-of-the-art large models for each step in our pipeline, these models are not 100% reliable and may produce bad examples. Considering we empolyed filtering strategies and BiAug is a unified framework that can evlove together with these models, BiAug is still usable, but the noise issue can be mitigated to further improve it. Utilizing large models such as LLMs and stable diffusion elevates the cost of data synthesis. Although this paper primarily addresses the efficacy of BiAug, future endeavors may alleviate the cost concern through methods including local deployment and parallel processing. While our pipeline employs state-of-the-art models, they do not guarantee absolute reliability and might occasionally yield suboptimal results. Although it remains effective with our employed filtering strategies and the adaptability of BiAug as a unified framework, addressing potential noise is important and will further enhance its performance.

## B  ETHICAL STATEMENT

We ensured that all resources used, including the dataset, benchmarks, and checkpoint of models, were accessed and utilized with respect to intellectual property rights and privacy concerns. No personally identifiable information was used. All implementations were designed to be transparent without the intention to produce new biases and ethical concerns.

## C  LLM PROMPT DESIGN

In this section we demonstrate the prompts we used in BiAug with 5-shot examples. For easy postprocessing, we ask LLM to generate candidate objects in the form of Python list. On the other hand, attributive description for objects and the augmented captions in the form of CSV format.

**Prompt 1: Object Candidates Proposal Prompt**

```
# SYSTEM MESSAGE
You are an experienced data annotator. All of your answer should be able to be directly
  transformed to a **python list**.

# HUMAN MESSAGE
Given an image caption. Infer **no more than five** possible objects that might be **
  visible** in the image. Answer in the form of python list, ranked in order of likelihood
  from most probable to least probable.

EXAMPLES:
Caption: "A vibrant carnival parade." -> Objects: ["Dancer in costume", "Float", "Spectator
  ", "Musician", "Street vendor"]
Caption: "An ancient temple ruin." -> Objects: ["Stone column", "Statue", "Broken wall", "
  Inscription", "Vegetation overgrowth"]
Caption: "A serene Japanese garden in spring." -> Objects: ["Koi fish", "Cherry blossom
  tree", "Stone lantern", "Wooden bridge", "Raked sand path"]
Caption: "An artist's studio." -> Objects: ["Paint brush", "Canvas", "Easel", "Palette", "
  Spilled paint tube"]
Caption: "A bustling train station." -> Objects: ["Commuter rushing", "Ticket machine", "
  Train track", "Digital departure board", "Bench seat"]

Caption: "{input_caption}" -> Objects:
```

**Prompt 2: Douple Object-Attribute Association Prompt**

```
# SYSTEM MESSAGE
You are an experienced data annotator. All of your answer should be able to be directly
  transformed to a **CSV file**. Now you are required to generate commonsense for each pair
   of objects and attributes based on the given caption, and generate new extended phrases
  for the objects, new extended captions for the image, **for training a commonsense-aware
  visual-language model**.

# Variables definition
[CAPTION]: a given description for an image.
[OBJECTS]: a list of given objects that are visible in the image.
```

```
[ATTRIBUTES]: ["color", "shape", "material", "other"]. These given attributes are used to
  describe the [OBJECTS].
[COMMONSENSE]: a commonsense concept that is not explicitly mentioned but can be inferred
  with commonsense based on the given [CAPTION].
[EXTENDED PHRASE]: a phrase that is extended from a [OBJECTS] with the generated [
  COMMONSENSE].
[EXTENDED CAPTION]: a caption that is extended from a [CAPTION] with the generated [
  EXTENDED PHRASE].
[NEGATIVE EXTENDED PHRASE]: a phrase, which should be distinguished from the [EXTENDED
  PHRASE] that is extended from a [OBJECTS] with another different [COMMONSENSE] in terms
  of corresponding [ATTRIBUTES].
[NEGATIVE EXTENDED CAPTION]: a caption that is extended from a [CAPTION] with the generated
   [NEGATIVE EXTENDED PHRASE].
[ANSWER]: a CSV file with six columns, in which the title is [OBJECTS], [ATTRIBUTES], [
  EXTENDED PHRASE], [EXTENDED CAPTION], [NEGATIVE EXTENDED PHRASE], [NEGATIVE EXTENDED
  CAPTION]. Columns are split by a comma. Each row a record for each pair of [OBJECTS] and
  [ATTRIBUTE].

Now, achieve the task step by step.
# Commonsense generation
Step one: infer [COMMONSENSE] for each pair of [OBJECTS] and [ATTRIBUTES]. If the generated
   [COMMONSENSE] is unclear, unknown, or not applicable, based on the [CAPTION], please
  skip the pair of [OBJECTS] and [ATTRIBUTES];
Step two: generate [EXTENDED PHRASE] for each [OBJECTS] based on each inferred [COMMONSENSE
  ];
Step three: generate [EXTENDED CAPTION] based on the [CAPTION] and the generated [EXTENDED
  PHRASE];
Step four: generate [NEGATIVE EXTENDED PHRASE] for each [OBJECTS] in terms of corresponding
   [ATTRIBUTES];
Step five: generate [NEGATIVE EXTENDED CAPTION] based on the [CAPTION] and the generated [
  NEGATIVE EXTENDED PHRASE];
# Answer in the required format
Finally, generate [ANSWER] in the format of CSV with **six columns**. **Title is required**
  , which should be [OBJECTS], [ATTRIBUTES], [EXTENDED PHRASE], [EXTENDED CAPTION], [
  NEGATIVE EXTENDED PHRASE], [NEGATIVE EXTENDED CAPTION].
Each row is a record each pair of [OBJECTS] and [ATTRIBUTE]. If a pair of [OBJECTS] and [
  ATTRIBUTE] is skipped, please skip the record for this pair of [OBJECTS] and [ATTRIBUTE]
  in the [ANSWER].

# Examples, if examples are empty, please skip this step
[CAPTION]: a very typical bus station
[OBJECTS]: ['bus']
[ANSWER]: [OBJECTS],[ATTRIBUTES],[EXTENDED PHRASE],[EXTENDED CAPTION],[NEGATIVE EXTENDED
  PHRASE],[NEGATIVE EXTENDED CAPTION]
bus,color,yellow bus,yellow buses at a very typical bus station,red bus,red buses at a very
   typical bus station
bus,shape,double-decker bus,double-decker bus at very typical bus station,single-deck bus,
  single-deck bus at a very typical bus station
bus,material,metal bus,metal bus at a very typical bus station,wooden bus,wooden bus at a
  very typical bus station
bus,other,city bus,city bus at a very typical bus station,school bus,school bus at a very
  typical bus station

[CAPTION]: a banana on the table
[OBJECTS]: ['banana', 'table']
[ANSWER]: [OBJECTS],[ATTRIBUTES],[EXTENDED PHRASE],[EXTENDED CAPTION],[NEGATIVE EXTENDED
  PHRASE],[NEGATIVE EXTENDED CAPTION]
banana,color,A yellow banana,a banana on the table is yellow,a green banana,a green banana
  on the table
banana,shape,A curved banana,a curved banana on the table,a straight banana,a straight
  banana on the table
banana,material,A ripe banana,a ripe banana on the table,an unripe banana,an unripe banana
  on the table
table,color,A wooden table,a banana on the wooden table,a metal table,a banana on the metal
   table
table,shape,A rectangular table,a banana on the rectangular table,a round table,a banana on
   the round table
table,material,A wooden table,a banana on the wooden table,a plastic table,a banana on the
  plastic table
table,other,A clean table,a banana on the clean table,a dirty table,a banana on the dirty
  table
```

```
[CAPTION]: salmon in the river
[OBJECTS]: ['salmon', 'river']
[ANSWER]: [OBJECTS],[ATTRIBUTES],[EXTENDED PHRASE],[EXTENDED CAPTION],[NEGATIVE EXTENDED
  PHRASE],[NEGATIVE EXTENDED CAPTION]
salmon,color,A pink salmon,salmon in the river is pink,a grey salmon,a grey salmon in the
  river
salmon,shape,A streamlined salmon,a streamlined salmon in the river,a bulky salmon,a bulky
  salmon in the river
salmon,material,A scaly salmon,a scaly salmon in the river,a smooth-skinned salmon,a smooth
  -skinned salmon in the river
salmon,other,an salmon fish,an alive salmon fish in the river,a slice of salmon food,a
  slice of salmon food in the river
river,color,A clear river,salmon in the clear river,a muddy river,salmon in the muddy river
river,shape,A winding river,salmon in the winding river,a straight river,salmon in the
  straight river
river,material,A rocky river,salmon in the rocky river,a sandy river,salmon in the sandy
  river
river,other,A fast-flowing river,salmon in the fast-flowing river,a slow-flowing river,
  salmon in the slow-flowing river

[CAPTION]: a cat lying on a carpet
[OBJECTS]: ['cat', 'carpet']
[ANSWER]: [OBJECTS],[ATTRIBUTES],[EXTENDED PHRASE],[EXTENDED CAPTION],[NEGATIVE EXTENDED
  PHRASE],[NEGATIVE EXTENDED CAPTION]
cat,color,A brown cat,a brown cat lying on a carpet,a black cat,a black cat lying on a
  carpet
cat,shape,A furry cat,a furry cat lying on a carpet,a hairless cat,a hairless cat lying on
  a carpet
cat,material,A fluffy cat,a fluffy cat lying on a carpet,a thin-haired cat,a thin-haired
  cat lying on a carpet
cat,other,A sleeping cat,a sleeping cat lying on a carpet,a awake cat,an awake cat lying on
   a carpet
carpet,color,A red carpet,a cat lying on a red carpet,a blue carpet,a cat lying on a blue
  carpet
carpet,shape,A rectangular carpet,a cat lying on a rectangular carpet,a round carpet,a cat
  lying on a round carpet
carpet,material,A woolen carpet,a cat lying on a woolen carpet,a synthetic carpet,a cat
  lying on a synthetic carpet
carpet,other,A soft carpet,a cat lying on a soft carpet,a hard carpet,a cat lying on a hard
   carpet

# Input
[CAPTION]: {input_caption}
[OBJECTS]: {input_objects}
[ANSWER]:
```

