# OpenReview forum: "Towards reporting bias in visual-language datasets: bimodal augmentation by decoupling object-attribute association"
_ICLR.cc/2024/Conference — Submitted to ICLR 2024_

### Official Review · Reviewer_chuK · 2023-10-27

**Soundness:** 2 fair
**Presentation:** 2 fair
**Contribution:** 2 fair
**Rating:** 5
**Confidence:** 4

**Summary:**

This paper studied the bias within V-L datasets and proposed a method to augment image-text pairs with multiple new emergent works. In detail, by decoupling the object and attribute in the images, locating the objects, adjusting the text, and inpainting the new images, this work rebuilt the V-L dataset in different scales and retrained baseline models. On several tasks, the proposed method is evaluated and compared with CLIP.

**Strengths:**

+ The bias within V-L training and problems of the datasets are important to dig.

+ The thought of using SOTA tools to augment the datasets is non-trivial. And the proposed pipeline looks sound.

**Weaknesses:**

- The whole paper lacks many details of method design choices, data curation, method, training, and inference details, thus hindering the readers to fully understand the effectiveness of the proposed method. Only empirical results are not enough.

- No bias analysis, which is the most important point in the introduction.

- Lacking experiments using different key tools, e.g., LLM, detector/grounder, image generator, etc, which may affect the data generation a lot.

- More solid analyses should be given to probe the relation between the data and the bias and performance of different tasks.

- Prompt: The font color is hard to read.

**Questions:**

1. I suggest the authors also discuss the problems in the causal view, like counterfactual samples.

2. Any user study of the image generation quality, prompt quality, and rationality of the V-l relations?

3. CLIP-ft: fine-tuning CLIP on the augmented data? Missing details.

---

> ### Author Response · Authors · 2023-11-22
>
> ```
> Q: The whole paper lacks many details of method design choices, data curation, method, training, and inference details, thus hindering the readers to fully understand the effectiveness of the proposed method. Only empirical results are not enough.
> ```
> A: Thanks for the valuable comment. We have reorganized and refined the paper from the following perspectives:
> * Update Figure 2 and the methodology with clearer presentation and more implementation details for the readers to better understand our approach.
> * Please refer to Section 5.1 for the training details.
> * Please refer to Section 4.1 for the implementation details including the data source and data processing.
> * We add human evaluation for each phases in this framework to better demonstrate the proposed method beyond the empirical results.
>
> ```
> Q: Prompt: The font color is hard to read.
> ```
> A: We have updated the prompt in Appendix C with carefully designed style for better readability.
>
> ```
> Q: No bias analysis, which is the most important point in the introduction.
> ```
> A: We want to clarify that reporting bias is the motivation of our method but not the exact problem we're studying in this paper.
>
> The proposed method is motivated by the phenomenon that object-attribute are usually associated (which is caused by reporting bias). This association could hinder the model to understand the attributes for an object, e.g. VL model could think _lemon_ is always _yellow_ because it sees that association frequently. In the proposed method, a _richer array of object-attribute_ pairs are added into the datasets **explictly**, thus it can help the VL model's ability to understand the object-attribute, instead of just memorizing the frequent patterns.
>
> ```
> Q: Lacking experiments using different key tools, e.g., LLM, detector/grounder, image generator, etc, which may affect the data generation a lot.
> ```
> A: Thank you for your suggestion! Reporting bias (as its property suggested) is often overlooked in existing models and datasets. As a result, it is non-trivel to reveal and fix it, hence requires multiple language/visual tools, making it hard to conduct large-scale experiments on a mass of combination of different foundation models. Hence, we use **state-of-the-art foundation models** to construct our framework, so that the generation quality could be as good as possible at the moment. We will leave the exploration of the effect of different foundation models in the future work to find the best practice for deployment.
>
> ```
> Any user study of the image generation quality, prompt quality, and rationality of the V-l relations?
> ```
>
> A: Thank you for the suggestion! We add a subsection for user study in Section 4.4 in the updated paper. Please refer to the paper for details.
>
> In summary, we did human evaluation for the foundation models outputs. We design five questions for the intermediate output:
> 1. Correct object: ''In step 1 and 2, does the framework generate reasonable objects?''
> 2. Implicit object: ''In step 1 and 2, do the valid objects contain the ones that are not explicitly mentioned in the original caption?''
> 3. Correct attribute: ''In step 3, does the framework add description for the correct given attributes?''
> 4. Valid negative: ''In step 4, can the pair of attributive objects serve as hard negative example to each other?''
> 5. Valid inpainting: ''In step 7, does the model inpatint the required object in the masked area?''
>
> It shows that we got >90\% correctness for _correct object_ and _valid negative_, around 60\% for _correct attribute_ and _valid inpainting_. And around 60\% augmented examples contains objects that are not explicitly mentioned in the original caption
>
> ```
> Q: More solid analyses should be given to probe the relation between the data and the bias and performance of different tasks.
> Q: I suggest the authors also discuss the problems in the causal view, like counterfactual samples.
> ```
> Thank you for your thoughtful feedback. As we mentioned above, our experiments involves extensive data processing and model training, which can be quite time-consuming. Due to the time limit, it may not be feasible for us to add more experimental analysis by the end of the discussion period. However, we will continue to enhance the analysis and discussion sections in the future.
>
> ```
> Q: CLIP-ft: fine-tuning CLIP on the augmented data? Missing details.
> ```
>
> A: CLIP-ft refers to fine-tuning on the source dataset sampled from CC3M but not on the augmented data. CLIP-ft is the baseline variant we compare with the proposed method BiAug. Specifically, we first sample dataset with different sizes from CC3M, i.e. the _source dataset_, then we apply the proposed method on the source dataset to obtain the _augmented dataset_. CLIP-ft is CLIP trained on the source one and BiAug is CLIP trained on the augmented one. They are comparable because the datasets are from the same source.
>
> We have modified the description in Section 5.1 to avoid misunderstanding.

---

> > ### Comment · Reviewer_chuK · 2023-11-23
> > **Post-rebuttal**
> >
> > Thanks for the response. After reading the reviews and responses, I tend to retain my initial rating. I think this paper could be revised and improved, especially the clarity of the complex pipeline design and experiments.

---

> > > ### Author Response · Authors · 2023-11-23
> > >
> > > Thanks for your reply! We have already thoroughly revised the manuscript based on reviewers' suggestion to make the pipeline clear. Modifications include rewriting the methodology, updating the pipeline figure2, improve the presentation of LLM prompts, and adding human evaluation for the intermediate outputs from the foundation models.
> > >
> > > Could you please kindly go through the updated manuscript? We believe the pipeline and experiments design have been improved to clear enough now. And don't hesitate to ask any specific questions requiring clarification!

---

### Official Review · Reviewer_NefH · 2023-10-31

**Soundness:** 3 good
**Presentation:** 4 excellent
**Contribution:** 3 good
**Rating:** 8
**Confidence:** 3

**Summary:**

This paper introduces an approach to address reporting bias in vision-language datasets. The proposed method, known as bimodal augmentation (BiAug), incorporates language models, including ChatGPT, GroundingDino, and Stable-diffusion-inpainting. These models are integrated to supplement the attribute information of objects in the datasets. Ultimately, this integration leads to the creation of a new vision-language dataset.

**Strengths:**

1. This paper stands out for its excellent writing and organization. Its clarity enhances the ease of comprehension and makes it highly accessible to readers.

2. The paper's motivation is not only intriguing but also holds significance within the context of vision-language tasks. The novelty lies in addressing the crucial issue of reporting bias. While it may appear that this paper seamlessly integrates large models, the rationale behind their usage is sound and nuanced, underpinning its innovative approach.

**Weaknesses:**

1. It appears that the current method focuses on data augmentation for vision-language (VL) datasets by altering one attribute of one object at a time. It would be beneficial if the author could elaborate on whether they have considered modifying multiple attributes and multiple objects simultaneously.

2. In addition to conventional attributes like color, shape, and material, has the author explored more detailed attributes, as in [1], provided by Large Language Models (LLM)?

3.  It is advisable for the authors to conduct a broader range of experiments and provide more visualization results, particularly in the context of Attribute Recognition Games (ARG) experiments.

4.  The meaning of the abscissa in Fig. 5 is not immediately clear. Further clarification or labeling may be needed for readers to grasp the content effectively. Furthermore, Fig. 5's subcaption would benefit from further adjustment.

5. The paper would benefit from a more extensive discussion of its limitations and potential directions for future research, offering a more comprehensive evaluation of the current method's scope and boundaries.

If the author can address my concerns, I would be willing to consider raising my scores.

[1] Sachit Menon and Carl Vondrick. Visual classification via description from large language models. arXiv preprint arXiv:2210.07183, 2022

**Questions:**

See `Weakness' above.

---

> ### Author Response · Authors · 2023-11-22
>
> Thanks for the time for reviewing our paper. Your comments and suggestions are very insightful! We're addressing your concerns as follows,
>
> ```
> Q: It would be beneficial if the author could elaborate on whether they have considered modifying multiple attributes and multiple objects simultaneously.
> ```
> A: This is a good point, especially we're processing a large scale of data. In fact, one attribute of one object at a time is just the example for presentation. We actually modify the all the four attributes for all valid objects to augment the captions, i.e. four attributes X multiple objects a time, please refer to Appendix C, prompt 2 for details.
>
> However, for the image inpaining, we do one attribute-object pair at a time, because modifying multiple objects could corrupt the image, and more importantly, we would not be able to construct the hard negative pairs to highlight the attributive difference for one object.
>
> ```
> Q: In addition to conventional attributes like color, shape, and material, has the author explored more detailed attributes
> ```
> A: This is also a very good point, we did explore different attributes but found it would be hard to cover every attribute in the large-scale experiments. To address the diversity of attributes, we add an **_other_** attribute to leverage the LLM’s generalization capabilities to infer relevant attributes from provided captions, highlighting the most relevant commonsense knowledge besides color, shape and material. Please refer to footnote 2 in the 4th page, and the prompt 2 in Appendix C.
> By doing this, LLM is able to generate attributive description dynamically based on the given caption, beyond the preset conventional attributes.
>
>
> ```
> Q: Add ARG experiments.
> ```
> A: Thanks for the suggestions! We're trying to add broader experiments by the end of discussion period if possible as the time limit. Could you also provide the reference or link to the ARG benchmark you mentioned? We'd also like to kindly note that we're focusing on the reporting bias issue (which is overlooked previously) in VL dataset, ARO is the most relevant benchmark to verify our ideas.
>
>
> ```
> Q: The meaning of the abscissa in Fig. 5 is not immediately clear. Further clarification or labeling may be needed for readers to grasp the content effectively. Furthermore, Fig. 5's subcaption would benefit from further adjustment.
> ```
> A: Thanks very much for the suggestion!
> The x-axis refers to the size of the source dataset used for training the model, e.g., 40K, 100K, 200K and 300K, for demonstrating the trend of model's performance with increasing data used for fine-tuning.
> We have added description for the x-axis in the caption of Figure 4 (Figure 5 in the original paper), and in Section 4.1 for better readability. As for the formatting in the figure, we'll carefully adjust them in the final version.
>
> ```
> Q: The paper would benefit from a more extensive discussion of its limitations and potential directions for future research, offering a more comprehensive evaluation of the current method's scope and boundaries.
> ```
> A: Thank you for the suggestion! We agree that discussion would be very helpful. We cannot add another section due to the page limit. Please refer to Appendix A for our discussion about limitation and future directions.

---

> ### Author Response · Authors · 2023-11-22
>
> We would like to express our sincere gratitude for the time and effort you have invested in reviewing our work. Your insights and comments have been invaluable in refining our manuscript.
>
> We are eager to learn your perspective on our responses to your comments. If there are any points or concerns that you believe have not been adequately addressed, please let us know and we are willing to provide further clarification and responses.

---

### Official Review · Reviewer_ANJ5 · 2023-11-05

**Soundness:** 3 good
**Presentation:** 3 good
**Contribution:** 2 fair
**Rating:** 6
**Confidence:** 5

**Summary:**

This paper focused on the problem of reporting bias in visual-language models. The authors proposed BiAug, a data augmentation framework through object-attribute decoupling. BiAug synthesizes vision-language examples and constructs hard negatives by utilizing foundation models including LLMs to extract objects, generate attribute descriptions and negative descriptions, a grounding object detector to detect objects, and an inpainting model to create new images. This paper compares with CLIP baselines on several object-attribute understanding benchmarks and zero-shot retrieval tasks. The results show the effectiveness of the proposed method.

**Strengths:**

1. The proposed BiAug framework utilizes several foundation models to generate synthetic and negative samples to disentangle objects and attributes. This is helpful to mitigate the reporting bias of VLMs.
2. The experiments on ARO dataset shows the effectiveness of proposed frameworks by comparing with two CLIP baselines. The experiments also show a clear trend of enlarging the dataset size, demonstrating the promising scalability of the proposed augmentation framework.

**Weaknesses:**

1. This paper utilizes multiple foundation models as black-box tools. It lacks of intuitive measurement of the quality of foundation models' output. For example, is there a numerical evaluation of the box quality generated in step 2 in Figure 2? In step 1 how is the quality of the "guessed" potential objects?
2. This paper claims also to achieve improvement in general text-image retrieval benchmarks. However, in figure 5 table (c) and (d), the performance of BiAug and Clip-ft is rather close on the 40k subset, and with more examples, BiAug didn't show clear improvement compared with either the BiAug or CLIP-ft baseline on the 40k subset. These results can't really show a solid improvement on the two datasets.
3. This work only compares with two baselines: CLIP and CLIP-ft. It is unclear whether BiAug can also benefit other VL models such as NegCLIP (Yuksekgonul et al., 2022) which has been trained with explicitly constructed negative samples.

**Questions:**

See Weakness section

---

> ### Author Response · Authors · 2023-11-22
>
> Thanks for your insightful comments! We're addressing your concerns as follows,
>
> ## Measurement of the quality of foundation models outputs
> That's a very good suggestion. So we add a subsection for that in Section 4.4 in the updated paper. Please refer to the paper for details.
>
> In summary, we did human evaluation for the foundation models outputs, as there is no benchmarks that are exactly designed for our case. To check whether these foundation models follow our instruction to work, we design five questions for the intermediate output:
> 1. Correct object: ''In step 1 and 2, does the framework generate reasonable objects?''
> 2. Implicit object: ''In step 1 and 2, do the valid objects contain the ones that are not explictly mentioned in the original caption?''
> 3. Correct attribute: ''In step 3, does the framework add description for the correct given attributes?''
> 4. Valid negative: ''In step 4, can the pair of attributive objects serve as hard negative example to each other?''
> 5. Valid inpainting: ''In step 7, does the model inpatint the required object in the masked area?''
>
> It shows that we got >90\% correctness for _correct object_ and _valid negative_, around 60\% for _correct attribute_ and _valid inpainting_. And around 60\% augmented examples contains objects that are not explicitly mentioned in the original caption
>
> As for your questions,
> ```
> is there a numerical evaluation of the box quality generated in step 2 in Figure 2?
> ```
> A: Grounding DINO achieves an impressive 60.7 mAP (refer to Table 2, [1]) in zero-shot object detection on the MSCOCO benchmark, without any prior training on this specific benchmark. This performance underscores its robust capability to reliably identify valid objects, which is essential for the effectiveness of our method.
>
> ```
> In step 1 how is the quality of the "guessed" potential objects?
> ```
> A: As the results shown Table 2 in the updated paper, the quality of the guessed potential objects are very good with a correctness >90\%.
>
>
> [1] Liu, Shilong et al. “Grounding DINO: Marrying DINO with Grounded Pre-Training for Open-Set Object Detection.” ArXiv abs/2303.05499 (2023): n. pag.
>
> ## Concern about the improvement on general VL retrieval task
> The Filcker30K-order and COCO-order in Figure 4(Figure 5 in older verion) are not the results for general text-image retrieval task, instead it's the results on a very challenging ARO benchmark[2].
> We have modified the paper to avoid misunderstanding.
> In ARO, the original Filcker and COCO captions are modified by changing partial word order, which are very similar to the original caption. This benchmark requires the VL model to pick the correct caption from a group of _hard negative_ examples. Our results show that BiAug has a more stable trend in distinguish _hard negative_ examples.
>
> The claim that BiAug also acheives improvement on general image-text retrieval task is supported by the results in Table 3. Please refer to the paper for details.
>
> [2] Yuksekgonul, Mert et al. “When and why vision-language models behave like bags-of-words, and what to do about it?” ArXiv abs/2210.01936 (2022): n. pag.
>
> ## Comparison with NegCLIP
> We're still reproducing the NegCLIP to make the comparison. We're afraid that it cannot be finished by the end of the discussion period as it requires a lot of time to preprocess the 300K data and train the models. We will update the results as soon as possible.
>
> However, we want to note that BiAug and NegCLIP are not actually comparable because our focus is to mitigate the reporting bias issue by decoupling the object-attribute association for **both visual and langauge modalities**, while NegCLIP just simply shuffle the caption, without making any changes to the objects or attributes in the caption and the image.

---

> > ### Comment · Reviewer_ANJ5 · 2023-11-22
> >
> > I'd like to thank the authors for the response and revision of the paper. I think these responses address my concerns. I'll make decision on my final scores after reading other rebuttal materials and discussing with other reviewers

---

> > > ### Author Response · Authors · 2023-11-22
> > >
> > > Thank you for the prompt reply! Don’t hesitate to ask if you have any further questions

---

### Official Review · Reviewer_6Azz · 2023-11-08

**Soundness:** 2 fair
**Presentation:** 1 poor
**Contribution:** 2 fair
**Rating:** 5
**Confidence:** 3

**Summary:**

**POST REBUTTAL NOTE FOR AUTHORS:**

I appreciate the effort put into this paper and acknowledge that I have read your responses.

-------------------------------------

**PRE REBUTTAL REVIEW:**

This paper focuses on the phenomenon of "reporting bias" in VLMs. "Reporting bias" refers to the case where part of the text description of an image is not present due to omission of implicit/commonsense knowledge. Authors claim that mitigating "reporting bias" may help with two issues: 1) biased captions 2) skewing the VLM towards frequently occurring patterns.

To mitigate the "reporting bias", authors propose "bi-modal augmentation" with the goal of disentanglement of the object-attribute associations. More specifically, BiAug augments a source dataset with explicit commonsense knowledge.

BiAug has three phases: 1) Cross-mode object extraction:identifies the objects that are not explicitly referred to in the caption, 2) Decouple object-attribute association: creates a pair of descriptions for the image and 3) Image synthesis: in-painting the image with the pair of descriptions found in phase (2).

The augmented datasets produced by BiAug are used to train models, which are then evaluated against benchmarks for object-attribute understanding and general vision-language retrieval tasks.

**Strengths:**

-  "Reporting bias" is an important concept that affects the performance of machine learning models in understanding and processing real-world data.

- The paper tackles the "reporting bias" problem and aims to provide a more nuanced dataset that includes objects and attributes often omitted due to reporting bias.

- This paper could have implications for AI fairness: By addressing reporting bias, the research moves towards creating AI systems that can potentially reduce cultural and linguistic biases, contributing to the broader goal of AI fairness and ethics.

Overall I found the problem to be very interesting and timely.

**Weaknesses:**

- I find the framework overly complicated and built on fragile assumptions. For example, why can we assume that the grounding object detector can find objects that a different color or shape of it is still a valid object? How can we make sure that the detected grounding object is big enough for the in-painting model to process this? Also see my questions below. I believe this overall complexity hinders the generalization of the results.

- Overall I did not find the paper easy to read. This is partly due to the fact that the methodology is very complex with lots of pieces. However, I think the authors need to spend more time thinking about the structure and also definitely re-writing the abstract.

- Many important concepts used in the paper are not clearly explained. For example "grounding object" need clear definitions.

- I find it hard to understand how the framework described in Figure 2 could be reliable for large scale experiments.

Minor points:

- I suggest having the term "reporting bias" italic or quoted as it is not a well-known term/concept. And it is not clear that it's a technical term. Specifically it is used in the beginning of the abstract as well as the introduction. I think both sentences in the start of the abstract and the start of the intro need to be re-written/clarified.

**Questions:**

- Where do the samples in Figure 1 come from? Are they from a real dataset? Are they generated?
- In Figure 2, in step 3, why "tennis ball" is dropped? It was part of the original text. Why "dog" is selected as the grounding object?
- In section 3.2 how do you select from the list of predefined categories? For example, in figure 2, how come the "shape" is not selected for "grass"? Of course it does not make sense to modify the shape for grass, but how do you account for that and select "color"?
- In 3.3 how do you mask the detected object?
- In the example given in figure 3, what happens if non of the predicted objects are inside the image?
- In figure 4, the "blue boat" and "red boat" samples, what was the missing reference object here according to your framework? What happened in the extraction phase? boat was already part of the original caption.
- What is "other" in the preset attribute list?

Overall, I found the figures and methodology very confusing. I am willing to modify my score based on the clarifications that the authors may provide. However, I am concerned with the overall writing and presentation of this work and I believe restructuring the paper to present the methodology in a more digestible and elaborate format could be very useful.

---

> ### Author Response · Authors · 2023-11-22
>
> ## We have refined the presentation in the updated version
> ```
> Q: However, I think the authors need to spend more time thinking about the structure and also definitely re-writing the abstract.
> ```
> Thank you for your insightful questions and suggestions. For your main concern about the presentation of this paper, we have carefully rewritten the abstract and method, updated the figure2, and updated the demonstration of prompt for LLM, for better readability. Please see the updated paper. Please also see our clarification below to your questions about the detail of the method.
>
> ```
> Q: Many important concepts used in the paper are not clearly explained. For example "grounding object" need clear definitions.
> ```
> A: Thank you for the suggestion. I went through the paper and add necessary background information for better readability.
>
> ## Concern about the relative complicated framework
> Reporting bias (as its property suggested) is often overlooked in existing models and datasets. As a result, it is non-trivel to reveal and fix it, such that our method consists of different important steps. For your specific concerns, we clarify them point by point as follows,
>
> ```
> Q: “why can we assume that the grounding object detector can find objects that a different color or shape of it is still a valid object?”
> ```
> A: It is important to emphasize that our system utilizes the state-of-the-art grounding detector, Grounding DINO [1], for detecting valid objects. This advanced detector is particularly efficient in recognizing objects based on textual input alone, even in a zero-shot context. Notably, Grounding DINO achieves an impressive 60.7 mAP (refer to Table 2, [1]) in zero-shot object detection on the MSCOCO benchmark, without any prior training on this specific benchmark. This performance underscores its robust capability to reliably identify valid objects, which is essential for the effectiveness of our method. Though GroundingDINO is generalizable, we agree that it is not perfect and could give mistakes. However, in our qualitative/human inspections in Section4.4, we found that GroundingDINO is generally reliable. And we expect that more advanced grounding models will be developed in the community and can be easily integrated to BiAug.
>
> ```
> Q: “How can we make sure that the detected grounding object is big enough for the in-painting model to process this?”
> ```
> A: We remove the detected boxes that are too large or too small to ensure they are valid for the following procedure. Please refer to the filtering strategies introduced in Section 4.1.
>
> ```
> Q: “How the framework described in Figure 2 could be reliable for large scale experiments.”
> ```
> A: We apply multiple filtering strategies, as introduced in Section 4.1, to remove low-quality grounding objects/augmented captions/generated images. Filtering strategies makes the examples we finally used for training are reliable as possible. We agree that we still cannot avoid noisy examples, even BiAug is utilizing state-of-the-art language and visual foundation models. The experimental results on general visual-language retrieval task shown in Table 3, also demonstrate that the dataset augmented by BiAug is reliable enough for training a VL model, as the performance keeps increasing with finetuning on the augmented dataset.
>
> [1] Liu, Shilong et al. “Grounding DINO: Marrying DINO with Grounded Pre-Training for Open-Set Object Detection.” ArXiv abs/2303.05499 (2023): n. pag.

---

> > ### Author Response · Authors · 2023-11-22
> > **following the previous response**
> >
> > ## Answers to other questions.
> > We have updated our paper and figure2, and added missing details for better presentation based on your suggestions. We’re answering your specific questions as follows.
> >
> > ```
> > Q1: Where do the samples in Figure 1 come from? Are they from a real dataset? Are they generated?
> > ```
> > A1: The “brown dog” example, which is also shown in Figure2, is from the CC3M that we used for our experiments, and the augmented two are generated by our framework. The other examples are collected from the internet for better conveying our motivation.
> >
> > ```
> > Q2: In Figure 2, in step 3, why "tennis ball" is dropped? It was part of the original text. Why "dog" is selected as the grounding object?
> > ```
> > A2: Some objects are kept, and others are removed based on our filtering strategies introduced in the last paragraph in Section 4.1. We remove too small and too large objects. We only keep objects that do not occupy over 70% of another object’s area, and a relatively higher grounding confidence to ensure the quality of examples as possible. In Figure 2, dog and grass are kept after filtering, and we take combination of “color” of “grass” as an example to illustrate the following steps.
> >
> > ```
> > Q3: In section 3.2 how do you select from the list of predefined categories? For example, in figure 2, how come the "shape" is not selected for "grass"? Of course it does not make sense to modify the shape for grass, but how do you account for that and select "color"?
> > ```
> > A3: “color” is just an example for illustration. We ask LLM to generate commonsense knowledge for all the four preset attributes, while an attribute-object pair will be skipped (which will be decided by the LLM as we require) if the object-attribute pair does not make sense. Please refer to Prompt 2 in Appendix C. Here we take a sentence in the prompt that we use to do this:
> >
> > _If the generated [COMMONSENSE] is unclear, unknown, or not applicable, based on the [CAPTION], please skip the pair of [OBJECTS] and [ATTRIBUTES];_
> >
> > ```
> > Q4: In 3.3 how do you mask the detected object?
> > ```
> > GroundingDino predicts the grounding box for the object, then we apply that box area as the mask for inpainting model. The stable-diffusion-inpainting takes an image, a mask, and a text prompt as input to edit the original image.
> >
> > ```
> > Q5: In the example given in figure 3, what happens if non of the predicted objects are inside the image?
> > ```
> > A5: In that case the grounding object detector will detect zero objects. As a result this example will be skipped because we get an empty list at Step 2.
> >
> > ```
> > Q6: In figure 4, the "blue boat" and "red boat" samples, what was the missing reference object here according to your framework? What happened in the extraction phase? boat was already part of the original caption.
> > ```
> > A6: In this example, we augment the original sample in terms of “missing attributive description” instead of “missing object”. Our proposed BiAug can augment the VL datasets in two folds: (1) complete the missing objects or missing attributes. (2) hard negative pairs guide the VL model learn to discern the attributive difference for identical object. Not all examples are applicable to all (but at least one) these perspectives.
> >
> > In the extraction phase, the object of ‘boat’ (but without the description in terms of color) was detected as explicitly mentioned objects. Subsequently, in the phase II, the boat object will be augmented in terms of the four preset attributes. For the case of color, it would be ‘red boat’ or ‘blue boat’.
> >
> > Please also refer to the table 2 in the updated paper, around 40\% of examples only contains valid object that are part of the original caption.
> >
> > ```
> > Q7: What is "other" in the preset attribute list?
> > ```
> > A7: It is the word “other”, in our prompt we ask LLM to infer commonsense concept for every object-attribute pairs, e.g., grass-color, grass-shape, grass-other. We add this to leverages the LLM’s generalization capabilities to infer relevant attributes from provided captions, highlighting the most relevant commonsense knowledge besides color, shape and material.

---

> ### Author Response · Authors · 2023-11-22
>
> We would like to express our sincere gratitude for the time and effort you have invested in reviewing our work. Your insights and comments have been invaluable in refining our manuscript.
>
> We are eager to learn your perspective on our responses to your comments. If there are any points or concerns that you believe have not been adequately addressed, please let us know and we are willing to provide further clarification and responses.

---

### Author Response · Authors · 2023-11-23
**General response to all reviewers**

We would like to express the gratitude for your insightful comments and suggestions on our manuscript.

We greatly value the reviewers' recognition of our work. Their acknowledgement highlights several key aspects:

* The importance of raising and addressing the reporting bias issue. (**6Azz**, "Reporting bias is an important concept..."; **chuK**, "
The bias within V-L training and problems of the datasets are important to dig." )
* The effectiveness of our proposed BiAug in mitigating the reporting bias, a crucial issue in VLM. (**ANJ5**, "...this is helpful to mitigate the reporting bias...", "...shows the effectiveness of proposed framework...", "...promising scalability...";)
* The broader implications of our work. (**6Azz**, "This paper could have implications for AI fairness...")
* The rational and sound usage of language/visual foundational models, which underscores the novelty of our proposed approach. (**NefH**, "...underpinning its innovative approach..."; **chuK**, "The thought of using SOTA tools to augment the datasets is non-trivial. And the proposed pipeline looks sound.")


Your feedback has been invaluable in improving the quality of our work. We have taken each of your points into consideration and made significant revisions to the paper as follows:

* In response to your concern about the complexity and readability of our method, we have restructured the methodology **Section 3** and updated **Figure 2** which illustrates our approach. We believe the revised version is much clearer and easier to understand.

* To address your curiosity about the practical effectiveness and reliability of the large models we utilized, we have added a **Section 4.4** on human evaluation. This new section thoroughly measures how existing language and visual foundation models work within our proposed framework and their performance.

* We have updated the presentation of the language model prompts in **Appendix C** to make it easier to understand how the language model works within the entire method. We believe this change will make the methodology more accessible to readers.

* We have also made targeted updates to areas of the paper where the expression was not clear, as per your other suggestions.

We believe these revisions have significantly improved the manuscript and hope you find the changes satisfactory. Once again, we appreciate your time and effort in reviewing our work.

---

### Meta-Review · Area_Chair_m8w8 · 2023-12-09

**Metareview:**

Four knowledgeable referees reviewed this submission. The reviewers raised concerns w.r.t.:
1. The presentation of the paper which could be significantly improved, in particular the motivation of the proposed approach and the method's presentation (chuK, 6Azz)
2. The assumptions made by the method and the generalizability of the presented framework (6Azz, ANJ5)
3. The unclear significance of the results (experiments appear rather limited, with missing baselines and improvements sometimes minor, a limited set of attributes is considered, and there is no bias analysis) (NefH, chuK, ANJ5)
4. The missing ablations on the effect of the choice of different tools as well as the failure modes of those (6Azz, chuK, ANJ5)
5. The limited discussion of limitations and future works (NefH)

The rebuttal partially addressed the reviewers concerns by clarifying the technical questions, refining the writing of some parts of the manuscript, introducing a discussion on limitations and future works and a human evaluation for the different foundational models. The rebuttal also clarified many of the questions raised by the reviewers. During discussion, and after reading the updated manuscript, the reviewers remain hesitant about the clarity of the presentation, and highlight discrepancies between the methodology described and the examples shown. The reviewers also discuss the importance of understanding what happens when one of the 6 modules in the framework fails, and the importance of including the missing comparisons. The AC agrees with the reviewers that not addressing these points hinders the significance of the contribution and the general applicability of the work. Therefore the AC recommends to reject but encourages the authors to introduce all the feedback provided by the reviewers to improve the next iteration of this work.

**Justification For Why Not Higher Score:**

Some of the main concerns raised by the reviewers on significance of results and clarity of presentation remain unaddressed during rebuttal. Addressing this feedback appears important to enable a better understanding of the potential of the presented contribution.

**Justification For Why Not Lower Score:**

N/A

---

### Decision · Program_Chairs · 2024-01-16

Reject